# Inequalities in Immunization against Maternal and Neonatal Tetanus: A Cross-Sectional Analysis of Protection at Birth Coverage Using Household Health Survey Data from 76 Countries

**DOI:** 10.3390/vaccines11040752

**Published:** 2023-03-29

**Authors:** Nicole E. Johns, Bianca O. Cata-Preta, Katherine Kirkby, Luisa Arroyave, Nicole Bergen, M. Carolina Danovaro-Holliday, Thiago M. Santos, Nasir Yusuf, Aluísio J. D. Barros, Ahmad Reza Hosseinpoor

**Affiliations:** 1Department of Data and Analytics, World Health Organization, 20 Avenue Appia, 1211 Geneva, Switzerland; 2International Center for Equity in Health, Federal University of Pelotas, Rua Mal Deodoro 1160, Pelotas 96020-220, Brazil; 3Department of Immunization, Vaccines, and Biologicals, World Health Organization, 20 Avenue Appia, 1211 Geneva, Switzerland

**Keywords:** inequality, maternal and neonatal tetanus, immunization, vaccination, health disparities

## Abstract

Substantial progress in maternal and neonatal tetanus elimination has been made in the past 40 years, with dramatic reductions in neonatal tetanus incidence and mortality. However, twelve countries have still not achieved maternal and neonatal tetanus elimination, and many countries that have achieved elimination do not meet key sustainability thresholds to ensure long-lasting elimination. As maternal and neonatal tetanus is a vaccine-preventable disease (with coverage of the infant conferred by maternal immunization during and prior to pregnancy), maternal tetanus immunization coverage is a key metric for monitoring progress towards, equity in, and sustainability of tetanus elimination. In this study, we examine inequalities in tetanus protection at birth, a measure of maternal immunization coverage, across 76 countries and four dimensions of inequality via disaggregated data and summary measures of inequality. We find that substantial inequalities in coverage exist for wealth (with lower coverage among poorer wealth quintiles), maternal age (with lower coverage among younger mothers), maternal education (with lower coverage among less educated mothers), and place of residence (with lower coverage in rural areas). Inequalities existed for all dimensions across low- and lower-middle-income countries, and across maternal education and place of residence across upper-middle-income countries. Though global coverage changed little over the time period 2001–2020, this obscured substantial heterogeneity across countries. Notably, several countries had substantial increases in coverage accompanied by decreases in inequality, highlighting the need for equity considerations in maternal and neonatal tetanus elimination and sustainability efforts.

## 1. Introduction

Maternal and neonatal tetanus (MNT) is a form of tetanus, an acute and potentially fatal disease caused by the bacterium *Clostridium tetani*. It affects women during pregnancy or within six weeks of the end of pregnancy and infants during their first 28 days of life [1]. MNT constitutes a major public health concern, as neonatal case-fatality rates are upwards of 80% and approach 100% when untreated [2]. Since the initial adoption of maternal and neonatal tetanus elimination (MNTE) goals by the World Health Organization (WHO) and global health partners in the late 1980s [3], the annual number of deaths due to neonatal tetanus has decreased substantially, from 787,000 in 1988 to 25,000 in 2018 [4]. MNTE, which is defined as less than one case of neonatal tetanus per 1000 live births in every district in a country each year (neonatal tetanus is considered a proxy indicator for maternal tetanus), has been achieved in 47 of the 59 priority countries targeted for MNTE as of December 2020 [5].

MNT is a vaccine-preventable disease [1,2]. Immunization is therefore a key strategy for achieving and sustaining its elimination, alongside clean birth and cord care practices, reliable surveillance, and use of data to identify areas and populations at risk for MNT [3]. To achieve life-long protection, the WHO recommends that national immunization programs provide six doses of tetanus toxoid containing vaccines (TTCV) administered in childhood and adolescence [2]. Pregnant women who are not vaccinated against tetanus, or for whom vaccination status is unknown, should receive at least two TTCV doses starting as early as possible during pregnancy. Pregnant women who are partially immunized with one to four doses should receive one dose before giving birth [2]. Thus, as populations increasingly receive the routine six doses during childhood and adolescence, fewer women will require TTCV during pregnancy.

MNT is associated with poverty and lack of access to adequate health services, and occurs most frequently in settings with weak health and immunization systems, largely in the worst performing districts in low- and lower-middle-income countries [1,2]. Therefore, MNT is inherently a health equity issue. Despite this, relatively few publications have examined predictors of and inequalities in maternal tetanus immunization, particularly relative to other child immunization outcomes. Prior research examining inequalities in childhood immunizations and using multi-national samples has found several factors which are significantly associated with disparities in coverage, including household wealth [6,7,8,9], maternal age [10], maternal education [8,9,11], and place of residence (urban/rural) [9,11,12,13]. A number of single-country studies have examined factors associated with tetanus vaccination uptake by pregnant women in, for example, Afghanistan [14], Bangladesh [15,16], Ethiopia [17,18], The Gambia [19], India [20], Kenya [21], Myanmar [22], Sierra Leone [23], and Sudan [24]. Across these studies, higher levels of maternal education and household wealth have often been found to be associated with increased TTCV uptake, and in some (but not all) contexts, there were also significant associations between uptake and maternal age and place of residence. Two multi-country studies within Africa found greater maternal age, education, and household wealth to be significantly associated with higher coverage of births protected against neonatal tetanus [25,26].

To date, no global multi-country analyses have explored the extent of inequalities in maternal tetanus immunization coverage. Though smaller-scale (e.g., country-level or subnational-area-level) analyses are important to understand context-specific determinants of maternal tetanus immunization coverage and inequalities, a multi-country examination such as this one provides the opportunity to assess whether broader trends in drivers of coverage and inequalities exist, by using consistent outcome and inequality dimension measures and methods. They also permit benchmarking (comparisons) between countries to identify different situations of inequality, and explore where lessons to address inequality can be learned or applied. Findings from multi-country analyses such as these are particularly useful for informing broad, multinational initiatives [27]. This study examines levels and trends in tetanus protection at birth by four dimensions of inequality (wealth, maternal age and education, and area of residence), and explores variations by country World Bank income level (low-, lower-middle-, and upper-middle income). Specifically, we hypothesize that factors shown to be associated with childhood immunization coverage (household wealth, maternal age, maternal education, and place of residence) will also be associated with MNT vaccination coverage across low- and middle-income country contexts. Quantifying and reporting inequalities in tetanus protection at birth can inform strategies and interventions to reach the goal of MNTE.

## 2. Materials and Methods

### 2.1. Data Sources

Data from this study come from 76 countries with a recent (2011–2020) Demographic and Health Survey (DHS) or Multiple Indicator Cluster Survey (MICS), which collected information on maternal tetanus immunization coverage during pregnancy [28,29]. DHS and MICS are nationally-representative household surveys that collect extensive information about health outcomes, interventions, and healthcare behaviors. The information analyzed here comes from interviews with women aged 15–49 years. DHS and MICS survey methodologies have been published elsewhere [30,31].

### 2.2. Study Outcome

We examine maternal tetanus immunization coverage via the standard measure Protection at Birth (PAB), the proportion of women whose most recent live birth was protected against neonatal tetanus [32,33]. A birth is considered protected from tetanus if the mother (a) received at least two doses of TTCV during the pregnancy for her most recent live birth; (b) received at least two doses of TTCV, the last one within 3 years of the most recent live birth; (c) received at least 3 doses of TTCV, the last one within 5 years of the most recent live birth; (d) received at least 4 doses of TTCV, the last one within 10 years of the most recent live birth; or (e) ever received at least 5 doses of TTCV at any time prior to the most recent live birth. This measure is based on women whose most recent live birth occurred in the 59 or 23 months prior to the survey for DHS or MICS, respectively. This difference in time frame is due to the data collection methodologies of the two survey families. Additionally, maternal tetanus vaccination is ascertained via recall in DHS, while maternal vaccination cards are requested for confirmation in MICS and recall is used only if no card is available.

### 2.3. Dimensions of Inequality

Based on drivers of inequality identified in previous publications on childhood vaccination as well as data availability in DHS and MICS, we examined the following four dimensions of inequality: household wealth (country-specific wealth quintiles) [6,7,8,9], maternal age (15–19, 20–49) [10], maternal education (none, primary, secondary or higher) [8,9,11], and place of residence (urban, rural) [9,11,12,13].

For a set of sub-analyses, we classified countries based on World Bank 2022 income groups: low-income, lower-middle income, or upper-middle income [34]. Only two high-income countries (Uruguay and Trinidad and Tobago) had available data, so they were excluded from these sub-analyses.

### 2.4. Statistical Analyses

We first present the latest situation of inequality in MNT vaccination coverage for each country (using the most recent survey available from 2011 to 2020) via disaggregated data and summary measures of inequality. For each of the four dimensions of inequality (household wealth, maternal age, maternal education, and place of residence), we calculated the following, based on the country-specific estimates:Median coverage by subgroup of each inequality dimension, overall;Median coverage by subgroup of each inequality dimension, by country income group;Absolute inequality in coverage between the most and least advantaged subgroups of each inequality dimension, calculated using difference (e.g., highest wealth quintile coverage minus lowest wealth quintile coverage) and the slope index of inequality (SII), overall and by country income group;Relative inequality in coverage between the most and least advantaged subgroups of each inequality dimensions, calculated using ratio (e.g., highest wealth quintile coverage divided by lowest wealth quintile coverage) and the relative index of inequality (RII), overall and by country income group.

For each median value estimated, we also present the 95% confidence interval (CI) of the median, calculated using the centile Stata command with default specifications. This uses a binomial-based method described in Mood and Graybill 1963 that makes no assumptions on the distribution of the coverage variable [35,36].

We also examined changes over time in coverage levels and coverage inequalities. For this analysis, we included countries with at least one survey in the period 2011–2020 and one in the period 2001–2010, where the two surveys were at least 5 years apart. When multiple surveys in a time range were available, the most recent survey that maintained a 5-year gap was used. To assess changes in inequality over time, we first examined annual absolute change in national coverage levels, calculated as the national coverage in the more recent survey minus the national coverage in the older survey divided by the number of years between surveys. We then calculated annual absolute excess change in coverage, which compares the annual rate of change in the least and most advantaged subgroups. This is calculated as (absolute annual change for least advantaged subgroup) minus (absolute annual change for most advantaged subgroup). Several patterns in coverage can lead to positive (pro-disadvantaged) or negative (pro-advantaged) excess change. For example, a positive excess change in coverage value can arise when both groups have increasing coverage but the increase in the disadvantaged group is faster than the increase in the advantaged group; when both groups have decreasing coverage but the decrease in the disadvantaged group is slower than the decrease in the advantage group; or when the disadvantage group increased (or had no change in) coverage while the advantage group had a decrease (or had no change in) coverage. Excess change in coverage has been previously used to portray change in inequality over time in diphtheria-tetanus-pertussis (DTP) immunization coverage and in other maternal health outcomes [7,37].

As an additional post hoc analysis, we examined whether trends in inequality differed substantially based on MNTE achievement status, assessing inequality metrics separately for countries who have achieved MNTE vs. those who have not achieved MNTE.

Relevant survey sampling designs were taken into account when calculating point estimates of disaggregated data and corresponding 95% CIs at the country level. Statistical significance was set at *p* < 0.05 for all comparisons, and 95% CIs are reported throughout.

We conducted all analyses in Stata 17, and we developed data visualizations using Tableau version 2022.1.1.

## 3. Results

### 3.1. PAB Coverage Medians

Median national PAB coverage among the most recent survey sample (N = 76) was 69.1% (95% CI 61.6–71.9%), ranging from 15.0% in Trinidad and Tobago to 91.8% in India. Median national PAB coverage was 71.0% (95% CI 58.2–79.3%) in low-income countries (*n* = 20), 71.3% (95% CI 66.1–76.1%) in lower-middle-income countries (*n* = 34), and 64.7% (95% CI 35.7–71.2%) in upper-middle-income countries (*n* = 20). PAB coverage differed by within-country populations subgroups for all examined dimensions of inequality (see Figure 1, Interactive Appendix A). Median PAB coverage increased monotonically with increasing wealth, from 61.6% (95% CI 51.4–70.2%) among the poorest quintile to 77.3% (95% CI 67.8–80.1%) among the richest. Children of younger mothers were less likely to have protection at birth, with median coverage increasing from 63.0% (95% CI 60.1–69.6%) among mothers aged 15–19 to 71.1% (95% CI 66.8–75.1%) among mothers aged 20–49. Maternal education was also associated with median PAB coverage, increasing from 63.1% (95% CI 52.5–69.2%) among mothers with no education, to 71.5% (95% CI 66.7–75.3%) among mothers with primary education, to 78.5% (95% CI 74.5–81.1%) among mothers with secondary or higher education. Finally, children in urban areas had higher median PAB coverage than children in rural areas (73.6% urban [95% CI 66.9–77.4%] vs. 66.0% rural [95% CI 59.1–72.7%]).

These patterns in inequalities in PAB coverage are largely consistent across country income groupings, with two exceptions. Increasing wealth and increased maternal age are not associated with increased coverage for upper-middle-income countries (see Figure 2a,b). Increasing wealth and older maternal age are associated with increased coverage for low-income and lower-middle-income countries, however. Within all three income groupings, we see a consistent increase in coverage with increasing maternal education (see Figure 2c) and greater coverage in urban compared to rural areas (see Figure 2d).

### 3.2. Absolute and Relative Inequality in PAB Coverage

The median difference between PAB coverage in the richest wealth quintile and poorest wealth quintile among the most recent survey sample was 7.9 percentage points (95% CI 5.0–11.8), and the ratio in coverage between these quintiles was 1.12 (95% CI 1.08–1.19) (see Table 1, Interactive Appendix A). These measures differed by country income grouping; in low-income countries, the median difference was 15.2 percentage points (95% CI 5.7–24.7), lower–middle-income countries had a smaller gap of 10.9 percentage points (95% CI 7.2–16.7), and upper-middle-income countries had a small negative gap of −1.5 percentage points in coverage (95% CI −6.5–5.5). The median ratio of coverage between the richest and poorest wealth quintile followed a similar pattern: 1.21 (95% CI 1.09–1.66) among low-income countries, 1.16 (95% CI 1.10–1.32) among lower-middle-income countries, and 0.98 (95% CI 0.88–1.08) among upper-middle-income countries.

Differences by maternal age were also evident. Overall, the median difference between PAB coverage in children of mothers aged 15–19 and mothers aged 20–49 among the most recent survey sample was 4.5 percentage points (95% CI 2.9–6.1), and the ratio in coverage between these groups was 1.07 (95% CI 1.04–1.10). In low-income countries, the median difference in coverage was 4.8 percentage points (95% CI 1.9–8.4). Lower-middle-income countries had a slightly larger gap of 6.0 percentage points (95% CI 3.7–8.3), while upper-middle-income countries had a small gap of 1.1 percentage points (95% CI −3.0–5.0) in coverage. The median ratio of coverage between the children of older and younger mothers followed a similar pattern: 1.09 (95% CI 1.03–1.18) among low-income countries, 1.09 (95% CI 1.06–1.14) among lower-middle-income countries, and 1.03 (95% CI 0.96–1.14) among upper-middle-income countries.

The median difference between PAB coverage in children of mothers with secondary or higher education and mothers with no education was 11.6 percentage points (95% CI 8.4–15.7), and the ratio in coverage between these groups was 1.18 (95% CI 1.12–1.25). Differences in PAB coverage by maternal education were substantial across all country income groups. In low-income countries, the median difference between PAB coverage in children of mothers with secondary or higher education and mothers with no education was 15.6 percentage points (95% CI 4.3–25.2). Lower-middle-income countries had a gap of 13.1 percentage points (95% CI 10.0–25.9), while upper-middle-income countries had a gap of 7.3 percentage points (95% CI −3.3–9.5). The median ratio of coverage between the children of more and less educated mothers followed a similar pattern: 1.23 (95% CI 1.05–1.53) among low-income countries, 1.20 (95% CI 1.17–1.47) among lower-middle-income countries, and 1.11 (95% CI 0.96–1.15) among upper-middle-income countries.

The median difference between PAB coverage in urban areas compared to rural areas was 3.7 percentage points (95% CI 1.9–6.4), and the ratio in coverage between these groups was 1.05 (95% CI 1.03–1.10). In low-income countries, the median difference in coverage was 7.3 percentage points (95% CI 3.8–11.3). Lower-middle-income countries had a gap of 4.7 percentage points (95% CI 1.3–8.1), while upper-middle-income countries had a negligible gap of 0.2 percentage points (95% CI −1.7–2.9). The median ratio of coverage between the children in urban versus rural areas followed a similar pattern: 1.10 (95% CI 1.05–1.21) among low-income countries, 1.07 (95% CI 1.02–1.13) among lower-middle-income countries, and 1.00 (95% CI 0.94–1.05) among upper-middle-income countries.

We analyzed both simple and complex measures of inequality in PAB coverage for each of the four examined dimensions of inequality. As simple and complex measures demonstrated similar patterns of results, we focus on reporting the simple measures of inequality (difference and ratio) here. Complex measure findings can be found in Interactive Appendix A.

### 3.3. Change in Inequality in PAB Coverage over Time

We focus our change over time results on inequalities in household wealth; findings for other dimensions of inequality are available in Interactive Appendix A). The change over time analyses included 41 countries with data in both the periods 2001–2010 and 2011–2020.

Examining annual absolute change in national average PAB coverage (see Figure 3, x-axis; Interactive Appendix A), we find almost no annual change (median −0.04 percentage points, 95% CI −0.35–0.76) in overall PAB coverage across the examined countries from earlier (2001–2010) to more recent (2011–2020) time frames. There is substantial variation by country, however, ranging from an annual decrease in coverage of 2.0 percentage points in Suriname to an annual increase of 2.6 percentage points in Afghanistan. Twelve countries saw annual improvements in coverage of 1 percentage point or more (suggesting at least a 10-percentage point improvement in coverage over the examined 10-year time period), while three countries saw annual decreases in coverage of at least 1 percentage point (suggesting at least a 10-percentage point decrease in coverage over the examined time period). Of note, no countries with 80% or higher coverage at the earlier time period (*n* = 8) saw any improvements in national coverage.

Examining annual absolute excess change in the poorest compared to the richest wealth quintiles (Figure 3, y-axis; Interactive Appendix A), we find an annual excess change median value of 0.26 percentage points (95% CI 0.05–0.41), indicating slightly more favorable change over time for the poorest quintile over the examined time period. This measure also demonstrated heterogeneity by country, ranging from 2.4 percentage points annual excess change in Liberia to −1.9 percentage points annual excess change in Zambia. Ten countries had excess annual change of 1 percentage point or more (equivalent to 10 percentage points or more over the examined time period, favoring the poorest quintile), while six countries had excess annual change of −1 percentage point or less (equivalent to 10 percentage points or more over the examined time period, favoring the richest quintile).

Six of the examined countries had a substantial increase in national average of 15 percentage points over the 10-year time period (annual change of 1.5 percentage points increase or more)—Afghanistan, Cambodia, Namibia, Nepal, Senegal, and Togo. All six countries also had positive annual absolute excess change, indicating faster improvement among the poorest than the richest. Afghanistan and Cambodia saw the largest statistically significant annual excess change, equivalent to 23 percentage points excess improvement for the poorest relative to the richest in Afghanistan, and 13 percentage points excess improvement for the poorest relative to the richest in Cambodia over the examined 10-year time period. The Gambia, Lesotho, Liberia, and Nigeria also indicated statistically significant excess change in favor of the poorest quintile, all four with excess change of 15 percentage points or more over the 10-year time period. Only three countries—the Democratic Republic of the Congo, Egypt, and Zambia—saw statistically significant excess change in favor of the richest quintile of 15 percentage points or more over the 10-year time period; all three countries saw decreases in their average national coverage over the same time period.

Examining the subset of 15 countries with data from the two most recent years of available data (2019–2020), we see substantial heterogeneity in change over time for coverage level and inequality by wealth quintile (see Figure 4). For example, Senegal had significant improvement in coverage levels for all wealth quintiles, but almost no changes in absolute inequality across levels of wealth. In contrast, Liberia had a negligible change in the national average coverage, but substantial reductions in inequality. Thus, while cross-national medians suggest little change for either coverage or inequality of PAB from 2001–2020, specific country patterns demonstrate meaningful changes over the time period.

### 3.4. PAB Coverage in Countries by MNTE Achievement Status

This study includes data from 10 of the 12 countries who have not achieved MNTE as of 2020: Afghanistan, Angola, Central African Republic, Guinea, Mali, Nigeria, Pakistan, Papua New Guinea, Sudan, and Yemen (Somalia and South Sudan have not met MNTE but did not have available data) [5]. All 10 countries demonstrated statistically significant inequality in PAB coverage across maternal education; nine had significant inequality in PAB across household wealth, nine had significant inequality in PAB across place of residence, and four had significant inequality in PAB across maternal age. Of the five countries for which we had data to examine change over time, three (Afghanistan, Nigeria, and Pakistan) demonstrated significant improvements in national average coverage over the examined time period, while two (Central African Republic and Mali) had stagnant coverage. Of these five, only Afghanistan and Nigeria had statistically significant excess change over time across any of the examined dimensions, indicating decreased inequality in PAB coverage by wealth in Afghanistan, and decreased inequality for all dimensions in Nigeria.

As a post hoc analysis, we also examined median inequality measures by MNTE achievement status (see Table 2, Appendix A). We find that there is substantially larger inequality (as measured by difference and ratio) in household wealth, maternal education, and place of residence among countries which have not achieved MNTE compared to those which have achieved MNTE; no meaningful difference in inequality by MNTE status is observed for maternal age.

## 4. Discussion

Findings from this study of 76 countries suggest that there is substantial inequality in maternal tetanus immunization coverage globally. In particular, we find substantial inequality in tetanus protection at birth coverage by household wealth quintile, maternal age, maternal education, and place of residence. Though previous studies have demonstrated inequalities in coverage in one or more of these inequality dimensions in single-country or single-continent contexts, this is the first study to examine inequalities in PAB coverage across all four of these dimensions utilizing a large, global sample of low-, lower-middle, and upper-middle income countries. As the burden of MNT is highest in the most vulnerable populations (including those with lower wealth, younger maternal age, lower maternal education, and rural residence) [1], the lower immunization coverage we observe in these groups is particularly concerning.

We find that greater maternal education and urban (compared to rural) residence are associated with greater PAB coverage, overall and for each country income grouping. This is consistent with prior research of other childhood vaccines, and with priority focus areas of major immunization initiatives, which include reducing gender-related barriers to immunization (such as maternal education) and reaching remote rural populations [12,27,38,39,40,41,42].

Older maternal age and higher household wealth are also associated with greater PAB coverage overall and for low- and lower-middle-income countries, similarly consistent with prior research and immunization targets. However, the upper-middle-income country group demonstrated approximately equitable coverage by maternal age and wealth. As MNTE has been achieved in all examined upper-middle-income countries, many for more than 20 years, tetanus toxoid vaccination efforts likely differ from those in low- or lower-middle-income countries, possibly resulting in alternate patterns of coverage [43].

With regard to age, upper-middle-income countries generally have higher and more equitable childhood vaccination coverage and have for the past several decades [7]—meaning more young mothers received the basic three doses of DTP vaccines and additional TTCV doses in childhood and adolescence, resulting in complete PAB coverage by the time of childbirth. We similarly expect inequalities in PAB coverage by maternal age to continue to narrow over time as childhood DTP3 and additional TTCV dose coverage increases. With regards to wealth, the fact that the lowest wealth quintile had the highest coverage was unexpected. Similarly, the observation of lower overall median PAB coverage across upper-middle-income countries (65%) compared to low- and lower-middle income countries (both 71%) was counter to hypothesized patterns based on other childhood vaccine coverages. These findings provide further evidence of differences in tetanus immunization strategies across country income groupings. This includes substantial supplementary immunization activities (SIAs) or campaigns in countries with the highest burden of maternal and neonatal tetanus, which are largely low- and lower-middle-income countries, and relatively few such activities in upper-middle-income settings [44]. Additionally, as MNTE has been achieved in all upper-middle-income countries analyzed, the disease is often no longer considered a priority public health issue, and immunization may be considered less necessary as there is near universal access to clean birth environments and adequate umbilical cord management practices [43]. Nonetheless, the findings regarding equitable PAB coverage by wealth within upper-middle-income countries, and relatively lower PAB coverage overall in these settings, warrant further exploration within country-specific contexts.

Differential patterns of PAB coverage across dimensions of inequality and country income grouping highlight the importance of examining multiple dimensions of inequality. However, this study examines only four potential factors which may influence PAB coverage. Additional factors, such as conflict-affected areas and intensity [24,45], or subpopulations defined by double disaggregation, such as urban poor [46,47], have been shown to be associated with lower PAB coverage. Future work using multi-country samples should examine these and other potentially related factors to better understand determinants of coverage levels and inequalities, and consider multivariate analyses to understand the relative importance of co-existing factors. For analyses of smaller geographic scope, examining factors which are as relevant and as specific to the context as possible will best enable targeted efforts to improve TTCV coverage and eliminate MNT [5].

Despite large strides in MNTE efforts over the examined time period, and success in achieving MNTE in 47 of 59 countries with MNT as of 2020, there has been little change in maternal tetanus immunization levels and inequalities over the study time period on aggregate. However, this hides significant heterogeneity in coverage levels and inequality across countries. We see significant improvements in PAB coverage of 15 percentage points or more over 10 years for six countries, all of which also demonstrated reductions in wealth-related inequality in PAB coverage over the same time period. Though we cannot determine the direction of this relationship in current analyses, efforts to improve coverage should simultaneously be oriented towards reducing inequality. Importantly, we see evidence of inequalities in PAB coverage for all ten examined countries which have not achieved MNTE, and only see improvements in coverage and inequality for two of these target countries. We also observe substantially greater inequalities in PAB coverage among countries which have not achieved MNTE compared to countries which have been successful in achieving MNTE for three of the four examined dimensions (wealth, maternal education, and place of residence) in the most recent data. Reductions in these inequalities in coverage will be crucial to achieve MNTE.

Efforts to improve PAB coverage and equity should thus remain a key aim of MNTE initiatives, including quality targeted supplementary immunization activities, increases in uptake by LMICs of TTCV booster doses along the life-course, improved antenatal care visit access and TTCV administration during antenatal care, and increased institutional deliveries and clean delivery practices [48,49,50,51]. Additionally, persistent inequalities in PAB coverage in those countries which have achieved MNTE suggest the need for ongoing efforts to ensure MNTE sustainability, such as periodic neonatal tetanus risk analyses and corrective measures to close immunity gaps [43]. Assessments of inequality such as this one may help inform the groups to be targeted in these MNTE sustainability efforts. Global initiatives, such as the Immunization Agenda 2030 (IA2030), also present opportunities to catalyze action to address inequalities in PAB [27]. Positioning maternal and neonatal tetanus as a tracer of inequality in health care provision will enable more visibility and enhanced resource mobilization for the global initiative to eliminate MNT.

This study relies largely on maternal vaccination self-report, which is subject to recall bias, particularly as childhood doses may have been received 20+ years prior [52]. No recent review has explored the reliability of recall for immunization coverage, but prior research suggests that it can be problematic for childhood vaccines [53]. In particular, older women and women with more children may be more susceptible to underreport prior doses, and maternal recall likely underestimates TTCV coverage generally [54]. However, TTCV immunization protocols indicate that women who do not remember if they received a dose—or who report that they have not received a dose—should be immunized; thus, successful TTCV immunization efforts should negate this bias. Increasing use of home-based records and digitalized personal health records will likely also lead to decreased recall bias, reduction in unnecessary doses, and improved coverage over time [55]. The complex nature of PAB definition requires surveyors to correctly and comprehensively collect information about past tetanus immunization, leading to potential underreporting of coverage if information is only partially collected. We do not have reason to think that such bias would differ by the dimensions of inequality examined, however. In particular, women with multiple prior pregnancies may have underreporting or inaccurate reporting of prior doses; limiting these analyses to first births only would help mitigate this potential bias. Though such analyses were outside of the scope of this manuscript, future work should consider examination of first births (single parity mothers) only. Despite limiting analyses to the most recent data available, we include surveys from 2011 to 2020, and the current situation in a country may have changed substantially in the time since. This is particularly a concern in light of the COVID-19 pandemic, which interrupted immunization efforts and healthcare access in many places. Conclusions from these analyses about specific country situations should therefore be interpreted with caution. Finally, the nature of this cross-sectional, aggregate analysis does not allow for conclusions about the relative importance of the examined dimension of inequality, a causal relationship between inequality and coverage levels, subnational inequalities in coverage, the relative contributions of immunization prior to versus during the most recent pregnancy, nor the most effective potential solutions for improving coverage and reducing inequalities. All of these areas would benefit from examination in future research.

Despite these limitations, findings from this work can be used to inform future research, policy, and clinical practice and to benchmark progress. The occurrence of maternal and neonatal tetanus is a marker of inequities as this disease affects the most vulnerable populations, thus, MNTE efforts should continue considering equity a priority to ensure sustained results. This includes regular data collection of PAB coverage along with sociodemographic data to be able to regularly perform disaggregated data monitoring and analysis. Findings from this routine monitoring then can and should be used to inform subpopulations which can be the targets of interventions to improve coverage including SIAs, additional ANC-based screening and vaccination opportunities, improved immunization documentation efforts, and tetanus awareness and education activities. These analyses also provide an initial set of potential priority groups (the lowest wealth quintile, lower maternal education, and rural populations) for vaccinations efforts, and provide a potential framework for identifying additional subpopulations of interest.

## 5. Conclusions

Maternal immunity against tetanus, measured as PAB coverage, is a key aspect of MNTE. Findings from this study of 76 countries suggest that substantial inequalities in PAB coverage exist for wealth, maternal age, maternal education, and place of residence, and that these inequalities exist globally across low-, lower-middle-, and upper-middle-income countries. Though global coverage changed little over the time period 2001–2020, several countries had substantial increases in PAB coverage accompanied by decreases in inequality, highlighting the need for continued equity considerations in MNTE efforts.

## Figures and Tables

**Figure 1 vaccines-11-00752-f001:**
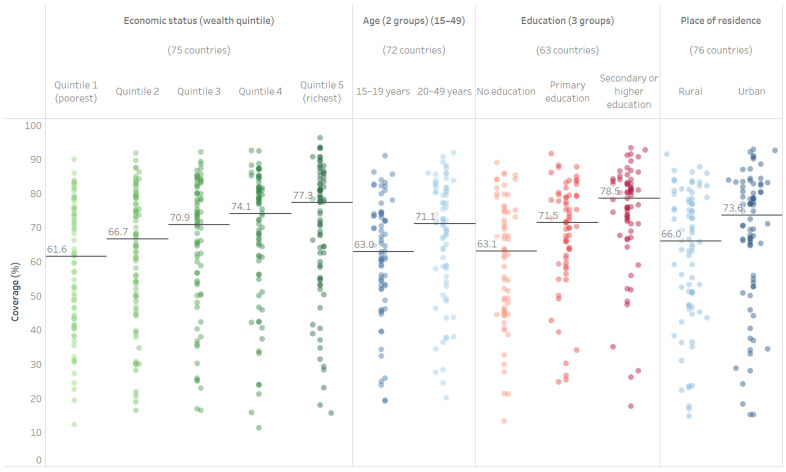
Latest situation of inequality in PAB coverage (DHS/MICS, 2011–2020).

**Figure 2 vaccines-11-00752-f002:**
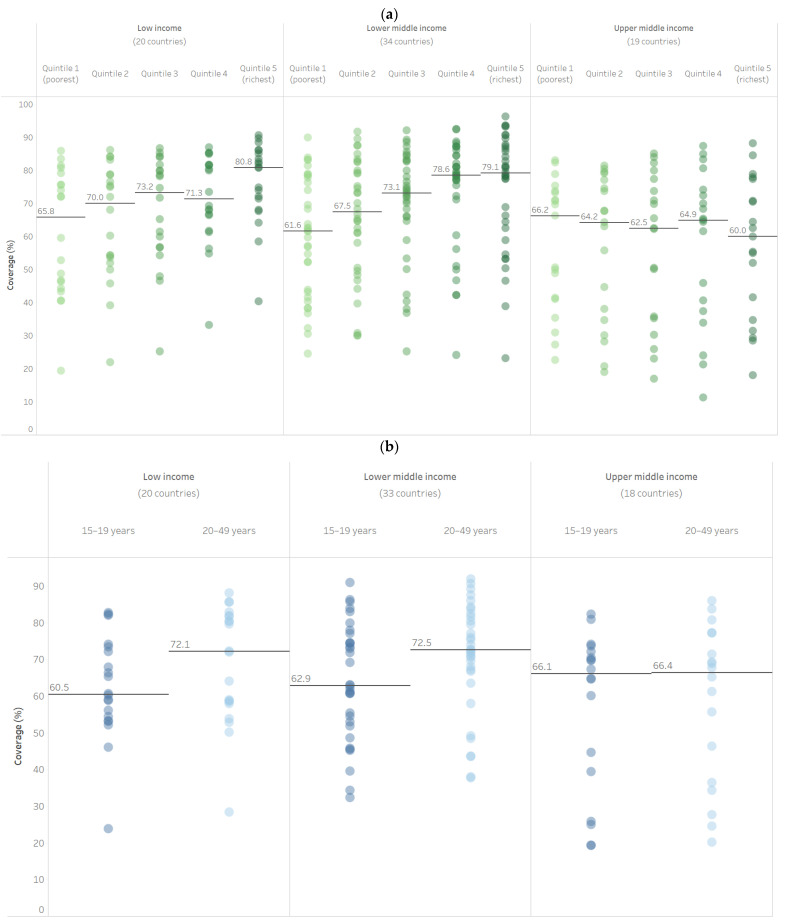
Latest situation of inequality in PAB coverage by World Bank income group (DHS/MICS, 2011–2020). (**a**) Wealth quintile; (**b**) maternal age; (**c**) maternal education; (**d**) place of residence.

**Figure 3 vaccines-11-00752-f003:**
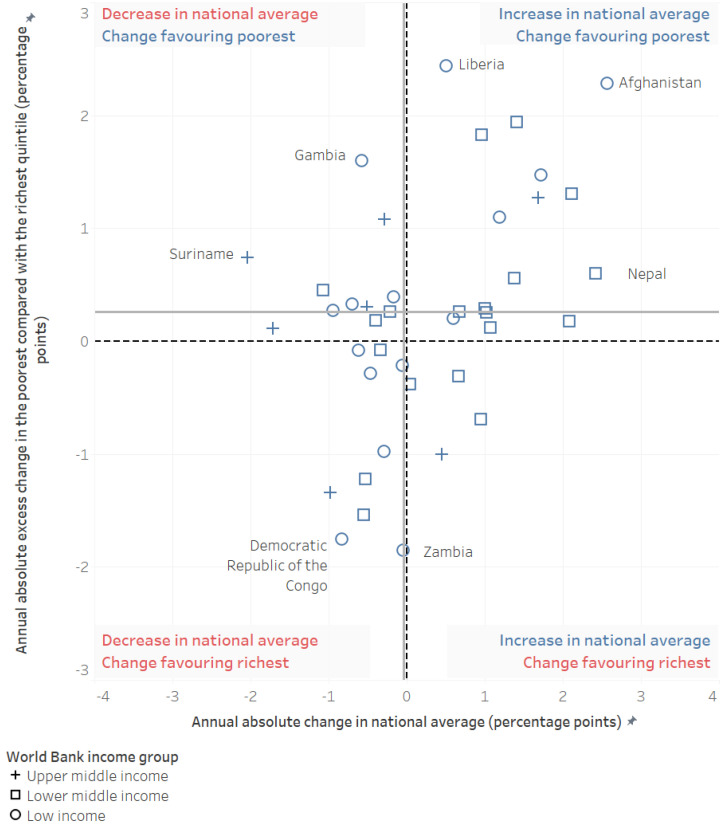
Change in national average and wealth-related inequality in PAB coverage (DHS/MICS, 2001–2010 and 2011–2020).

**Figure 4 vaccines-11-00752-f004:**
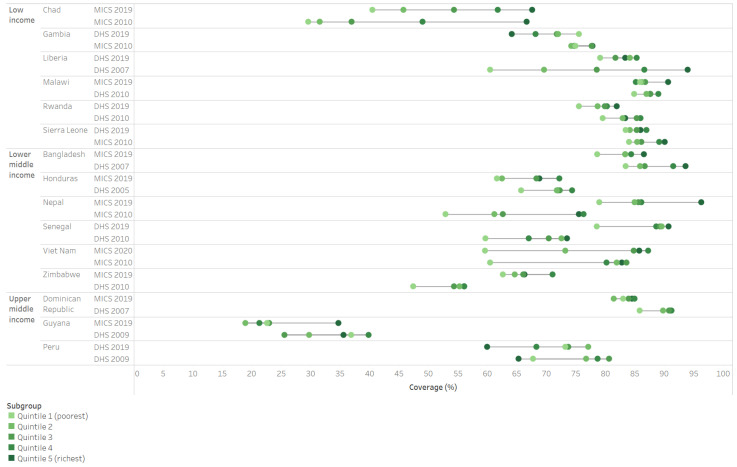
Change in inequality in PAB coverage (DHS/MICS, 2001–2010 and 2011–2020), countries with latest survey in 2019 or later.

**Table 1 vaccines-11-00752-t001:** Median difference and ratio in PAB coverage across four dimensions of inequality, overall and by World Bank income group (DHS/MICS, 2011–2020).

	Household Wealth Highest vs. Lowest Quintile		Maternal Age Age 20–49 vs. Age 15–19		Maternal Education Secondary or More vs. No Schooling		Place of Residence Urban vs. Rural	
	Difference (95% CI)	Ratio (95% CI)	Difference (95% CI)	Ratio (95% CI)	Difference (95% CI)	Ratio (95% CI)	Difference (95% CI)	Ratio (95% CI)
All Countries	7.9 (5.0–11.8)	1.12 (1.08–1.19)	4.5 (2.9–6.1)	1.07 (1.04–1.10)	11.6 (8.4–15.7)	1.18 (1.12–1.25)	3.7 (1.9–6.4)	1.05 (1.03–1.10)
Low-Income Countries	15.2 (5.7–24.7)	1.21 (1.09–1.66)	4.8 (1.9–8.4)	1.09 (1.03–1.18)	15.6 (4.3–25.2)	1.23 (1.05–1.53)	7.3 (3.8–11.3)	1.10 (1.05–1.21)
Lower-Middle Income Countries	10.9 (7.2–16.7)	1.16 (1.10–1.32)	6.0 (3.7–8.3)	1.09 (1.06–1.14)	13.1 (10.0–25.9)	1.20 (1.17–1.47)	4.7 (1.3–8.1)	1.07 (1.02–1.13)
Upper-Middle Income Countries	−1.5 (−6.5–5.5)	0.98 (0.88–1.08)	1.1 (−3.0–5.0)	1.03 (0.96–1.14)	7.3 (−3.3–9.5)	1.11 (0.96–1.15)	0.2 (−1.7–2.9)	1.00 (0.94–1.05)

**Table 2 vaccines-11-00752-t002:** Median difference and ratio in PAB coverage across four dimensions of inequality, overall and by MNTE status (DHS/MICS, 2011–2020).

	Household Wealth Highest vs. Lowest Quintile		Maternal Age Age 20–49 vs. Age 15–19		Maternal Education Secondary or More vs. No Schooling		Place of Residence Urban vs. Rural	
	Difference (95% CI)	Ratio (95% CI)	Difference (95% CI)	Ratio (95% CI)	Difference (95% CI)	Ratio (95% CI)	Difference (95% CI)	Ratio (95% CI)
All Countries	7.9 (5.0–11.8)	1.12 (1.08–1.19)	4.5 (2.9–6.1)	1.07 (1.04–1.10)	11.6 (8.4–15.7)	1.18 (1.12–1.25)	3.7 (1.9–6.4)	1.05 (1.03–1.10)
Countries which have NOT achieved MNTE	30.6 (23.1–46.1)	1.89 (1.67–2.14)	2.0 (−1.7–8.0)	1.04 (0.97–1.17)	28.4 (23.8–37.4)	1.67 (1.52–2.17)	18.0 (11.3–26.0)	1.40 (1.21–1.56)
Countries which have achieved MNTE	6.4 (3.5–9.8)	1.10 (1.05–1.15)	4.7 (3.0–6.2)	1.07 (1.05–1.10)	9.1 (5.6–12.0)	1.15 (1.08–1.19)	3.0 (1.4–5.0)	1.04 (1.02–1.07)

## Data Availability

All analyses were carried out using publicly available datasets that can be obtained directly from the DHS (dhsprogram.com) and the MICS (mics.unicef.org) websites. Data access is free and publicly available, but requires registration as a data user with the corresponding survey organization and submission of a request to access data for research purposes (https://dhsprogram.com/data/Access-Instructions.cfm; https://mics.unicef.org/visitors/sign-up). Datasets are continuously sourced and updated by the International Center for Equity in Health (equidade.org) as they are released. Analyses used the latest available dataset versions as of 18 December 2022.

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
