# Peer review of "Inequalities in Immunization against Maternal and Neonatal Tetanus: A Cross-Sectional Analysis of Protection at Birth Coverage Using Household Health Survey Data from 76 Countries"

_vaccines, 2023, doi:10.3390/vaccines11040752_

Round 1

Reviewer 1 Report

Overall a well-written and thought-provoking manuscript. Authors also did a good description of study design. The paper is suitable for publication but there are a few issues that need to be addressed. Figures are not readable due to bad quality. More important, a paragraph describing how the findings of this study can guide further research/clinical work would be useful.

Author Response

The authors thank Reviewer 1. Please see attached file for response.

Reviewer 2 Report

The article is interesting because tetanus is a preventable disease that can cause a painful death. From the methodological point of view, the approach is correct and supported by updated bibliography.

The following changes are suggested.

 1) Please include the link to the survey. Demographic 90 and Health Survey (DHS) or Multiple Indicator Cluster Survey (MICS),

2) In the text, the authors state, "Based on drivers of inequality identified in previous publications on childhood 113 vaccination as well as data availability in DHS and MICS, we examined the following four 114" please provide bibliographic references to support this statement.

3) In the material and methods, the authors indicate that they used a binomial distribution to calculate the median 95% Confidence interval. They should explain why they have used this method instead of  other methods, such as the normal approximation method, the bootstrap method, the Hodges-Lehmann estimator method, the rank-based method, or the Bayesian method.

4) Please comment that further research is needed into how different population groups can best access vaccines to ensure equitable immunization rates across all demographics.

5) Finally, the comment would also benefit future studies to examine other factors, such as gender or ethnicity, which may influence vaccine uptake and lead to disparities between populations.

6) Please comment on the policy implications of this paper. Some suggestions are provided below.

+ maternal and neonatal tetanus elimination (MNTE) efforts should consider equity to ensure long-lasting results

+ This study found substantial inequalities in protection at birth, a measure of maternal immunization coverage, across 76 countries for wealth, age, education level, and residence. • These findings suggest that MNTE programs need to focus on providing equitable access to vaccines so as not only to increase overall coverage but also to reduce the existing disparities between different population groups.

+  Additionally, continued monitoring is needed over time since global PAB coverage changed little from 2001-2020 despite some countries having significant increases accompanied by decreases in inequality during this period

7) This paper could be improved by providing examples or more detailed information on the specific interventions implemented in countries to reduce inequalities and increase coverage.

Author Response

The authors thank Reviewer 2. Please see attached file for response.

Reviewer 3 Report

Thank you very much for sharing this article with me. This is a well written piece and I have several minor comments.

1. In Introduction, please briefly discuss the importance of multi-country analyses. What is the motivation of conducting this study? What is the advantage of this approach? How does this approach contribute to the literature theoretically and methodologically? 

2. Is it important to extend the analysis to include parity (i.e., number of birth), which is further linked to women's empowerment (however defined)?

3. I'd like to see if the trend of inequalities differs based on countries with MNTE and those without it. Is it possible/important to highlight this?

Author Response

The authors thank Reviewer 3. Please see attached file for response.
